# Virulence profiles of some *Pseudomonas aeruginosa* clinical isolates and their association with the suppression of *Candida* growth in polymicrobial infections

Rehab Mahmoud Abd El-Baky[1,2]*, Sahar A. Mandour[2], Eman Farouk Ahmed[2], Zeinab Shawky Hashem[1], Tim Sandle[3], Doaa Safwat Mohamed[2]

1 Department of Microbiology and Immunology, Faculty of Pharmacy, Minia University, Minia, Egypt,
2 Department of Microbiology and Immunology, Faculty of Pharmacy, Deraya University, Minia, Egypt,
3 School of Health Sciences, Division of Pharmacy & Optometry, University of Manchester, Manchester, United Kingdom

* dr_rehab010@yahoo.com, rehab.mahmoud@mu.edu.eg

## Abstract

*Pseudomonas aeruginosa* is an opportunistic pathogen that can cause a variety of diseases especially in the hospital environment. However, this pathogen also exhibits antimicrobial activity against Gram-positive bacteria and fungi. This study aimed to characterize different virulence factors, secreted metabolites and to study their role in the suppression of *Candida* growth. Fifteen *P. aeruginosa* isolates were tested for their anticandidal activity against 3 different *Candida* spp. by the cross-streak method. The effect on hyphae production was tested microscopically using light and scanning electron microscopy (SEM). Polymerase chain reaction was used in the detection of some virulence genes. Lipopolysaccharide profile was performed using SDS-polyacrylamide gel stained with silver. Fatty acids were analyzed by GC-MS as methyl ester derivatives. It was found that 5 *P. aeruginosa* isolates inhibited all tested *Candida* spp. (50–100% inhibition), one isolate inhibited *C. glabrata* only and 3 isolates showed no activity against the tested *Candida* spp. The *P. aeruginosa* isolates inhibiting all *Candida* spp. were positive for all virulence genes. GC-Ms analysis revealed that isolates with high anticandidal activity showed spectra for several compounds, each known for their antifungal activity in comparison to those with low or no anticandidal activity. Hence, clinical isolates of *P. aeruginosa* showed *Candida* species-specific interactions by different means, giving rise to the importance of studying microbial interaction in polymicrobial infections and their contribution to causing disease.

## Introduction

*Pseudomonas aeruginosa* is an opportunistic Gram-negative organism that is widely distributed in the environment, including the hospital environment and the bacterium is part of normal flora of healthy individuals and immunocompromised hosts. In the hospital setting, *Pseudomonas* spp. may be obtained from ventilators and other respiratory equipment, sinks

**Data Availability Statement:** All relevant data are within the paper and its Supporting Information files.

**Funding:** The authors received no financial support for the research or publication of this article.

**Competing interests:** The authors have declared that no competing interests exist.

and toilets. The bacterium can cause a variety of infections that range from mild to life-threatening infections, and because of its connection to nosocomial environments, it poses a particular risk to immunocompromised patients [1].

*Pseudomonas* spp. possesses a panel of virulence factors that play key role in their invasiveness, toxicity, pathogenicity, and their ability to evade the host immune system. In addition, Pili, exoenzyme S, *LasB* elastase, exotoxin A and Phospholipase C virulence factors are involved in the more acute infections caused by *Pseudomonas* spp., while alginate and siderophores are involved in the triggering of chronic infections [1].

Many researchers have reported the antimicrobial activity of *Pseudomonas* spp. against fungi, Gram positive cocci and Gram-negative diplococci but not species of Gram-negative bacilli. This inhibitory activity is attributed to secretion of compounds with antimicrobial activity such as phenazines, pyrrolnitrin, and pseudomonic acid [2, 3]. Also, many studies have demonstrated the effectivity of Pseudomonads as biocontrol agents due to their inhibitory activity against soil microbes and fungi, suggesting that the antimicrobial properties can be deployed as biological pesticides in place of organic or inorganic chemical pesticides for use in agriculture [4, 5]

*P. aeruginosa* plays an important role in cystic fibrosis infection and infections associated with burns; here it can be co-isolated with *Candida albicans* [6]. It has been found that *Pseudomonas* spp. can inhibit the growth of many fungi such as *Cryptococcus* spp. and *C. albicans* [7–10]. Colonization by *Pseudomonas* spp. and *C. albicans* is regulated by selective pressure exerted by low nutrient availability and bacterial-fungal competition. When equilibrium is disrupted, opportunistic infections arise [11]. *P. aeruginosa* reacts with *C. albicans* antagonistically by killing *C. albicans* hyphal cells through alteration of cell wall via the action of pyocyanin [12] or reversion of the formation of germ tubes. Resistance of *C. albicans* yeast cells against the killing effect of *P. aeruginosa* is possibly connected to O-linked mannans [9, 13].

Fifty natural phenazine compounds are produced by *Pseudomonads* [14, 15]. Phenazines are colorful, nitrogen—containing products possessing antifungal and antibacterial activity [16]. One of the most toxic phenazines is 5-methyl-phenzaine-1-carboxylic acid, which shows greater toxicity compared with pyocyanine (5-methylphenazine-1-one). Furthermore, studies show that fungal death can be induced by 5-methyl-phenazine-1-carboxylic acid more efficiently than PCA (phenazine-1-carboxylic acid) and pyocyanine when *C. albicans* and *P. aeruginosa* are cultured together [7, 17].

Phenazines such as pyocyanine increase the virulence of *P. aeruginosa* by impairing the engulfing action of apoptotic cells, respiration of mammalian cells, and beating of human cilia in the respiratory system [18, 19]. Furthermore, it has been found that virulence decreases in MDR *P. aeruginosa* by decreasing pyocyanin production [17, 20]. The inability to produce pyocyanin lowers the pathogenicity of *Pseudomonas* spp. which is also found to be highly susceptible to immune defense [21]. *Pseudomonas*-competing microorganisms are eliminated by the antimicrobial effect of pyocyanin; this provides a significant advantage that helps *Pseudomonas* organisms to predominate in co-cultures [22–25].

Other studies show that quorum sensing controls interaction between *P. aeruginosa* and *C. albicans* where N-acyl homoserine lactone plays a sensory role for induction of target site expression [26]. It is also reported that production of pseudomonal surface adherence proteins is dependent on 3-oxo-HSL (3-oxododecanoyl- L-homoserine lactone) as this autoinducer is required for Pseudomonal adherence to *Candida* hypha [27].

This work studied the anticandidal activity and the virulence profile of fifteen *P. aeruginosa* isolates, the distribution of virulence factors among *P. aeruginosa*, and their association with anticandidal activity.

## Methods and materials

### Isolation and identification

One hundred fifty samples were collected from intensive care unit (ICU) patients suffering from different infections (septicemia, pulmonary, wound and urinary tract infections) as part of the routine hospital laboratory procedures. Samples were processed and cultured using traditional microbiological procedures to isolate and identify isolates. It was found that out of these 150 samples, 50 samples were positive for *P. aeruginosa* and 8 samples were positive for different *Candida* spp. Co-existence of *P. aeruginosa* and the isolated *Candida* was observed in 6 samples (4 skin swab and 2 urine samples). Complete identification to the isolated strains was undertaken as follows:

*P. aeruginosa* isolates were identified by Gram-stain, colony morphology, and ability to grow on cetrimide agar at 42˚C. The isolated colonies were tested for their biochemical characteristics (catalase, oxidase, triple sugar iron, urease, and sulfide indole motility). Colonies were purified by streaking and stored at—80˚C in stocks with 2.5 M glycerol. We selected 15 *P. aeruginosa* of which 6 isolates ($P_5$, $P_8$, $P_9$, $P_{16}$, $P_{100}$ and $P_{85}$) were isolated in association with the *Candida* isolates to be incorporated in this study.

*Candida* strains were cultured on Sabouraud dextrose agar (Lab M, UK). Eight *Candida* strains were identified by macroscopic features (morphology, color, size, and texture) on Sabouraud's dextrose agar (SDA) and CHROMagar Candida (CHROMagar Candida, France) showing 6 *C. albicans* and 2 *C. krusei*. (*C. albicans* isolated from samples positive for *P. aeruginosa* were $C_5$, $C_9$, $C_{16}$ and $C_{100}$ while *C. krusei* isolated from samples positive for *P. aeruginosa* were $C_8$ and $C_{85}$). Two *C. glabrata* ($C_{T1}$ and $C_{T9}$) were obtained from the department of Microbiology and Immunology, Faculty of Pharmacy, Minia University to be included in our study. Preservation of isolates were maintained at −70˚C in Trypticase Soya Broth (TSB, Becton and Dickinson) with 10% glycerol.

### Cross-streak method

*P. aeruginosa* strains were incubated overnight at 37˚C in order to prepare fresh cultures; from these, an inoculum was prepared in 0.9% NaCl. *P. aeruginosa* strains were cross streaked at 1cm width with sterile cotton swab on Sabouraud dextrose agar and incubated for 24 hrs. After overnight incubation at 37˚C. *P. aeruginosa* growth was removed by sterile glass slide. In a safety cabinet, sterile filter paper soaked in chloroform was placed in the cover of petri dishes from which the cultures of *P. aeruginosa* were removed with the sterile slides. Then, plates were covered and inverted to the face containing chloroform filter papers for 30 minutes in order to kill the microscopic cultural remnants. After 30 minutes, chloroform filter papers were removed, and plates were exposed to air for a few minutes to eliminate chloroform traces. Freshly prepared *Candida* cultures were streaked onto *P. aeruginosa-Sabouraud Dextrose* Agar plates. Streaking was performed perpendicularly to the original line of *P. aeruginosa* culture then plates were incubated at 37˚C overnight. Plates were tested for *Candida* growth inhibition as follows: -, no inhibition; +, 25% inhibition; ++, 50% inhibition; +++, 75% inhibition; ++++, 100% inhibition of *Candida* culture line [3]. Blood agar culture media is also used with the same previous procedure in Kerr's method in order to detect the anticandidal activity of *P. aeruginosa* strains [28].

### The effect of the tested *P. aeruginosa* on the morphology of *Candida* spp.

**Using light microscope to test the effect of cell-free filtrate of *P. aeruginosa* on the growth of candida cells.** *P. aeruginosa* isolates showing anticandidal activity were inoculated into 10 ml of Trypticase soya broth (TSB) and incubated at 37˚C for 2 days. Then, centrifuged at 10000x g for 15 minutes. Supernatants were exposed to filtration using sterile bacterial filters

0.2 μm. The filtrate was kept at 4˚C until use. Fresh overnight culture suspension (20 μl) of the tested strains of *C. albicans* and *C. krusei* (each alone) was added to half ml of fresh human serum (peptone water, human plasma, or TSB can be also used for germ tube production) that was distributed in 7 tubes for each of the tested *Candida* (*C. glabrata* does not form germ tube). Cell free filtrate of the 15 *P. aeruginosa* isolates were added to tubes. One tube for each *Candida* isolate was used as a control containing human serum inoculated with *Candida albicans* or *C. krusei* without adding *P. aeruginosa* cell free filtrate. Tubes were tested for germ tube formation using light microscope (Lecia, Germany) after 3 hrs incubation at 37˚C.

**Using Scanning Electron Microscope (SEM).** *P. aeruginosa* isolates showing anticandidal activity were co-cultured with *C. albicans* in trypticase soya broth (TSB) at 37˚C for 48 hrs. Then, samples were prepared for SEM. Cells were fixed in 2.5% glutaraldehyde for 1 hour at 4˚C, followed by rinsing cells with phosphate buffer saline (PBS). Series of ethanol drying was performed. Samples were coated with gold using cathodic spraying. Finally, examination was performed using JSM-840 SEM (JEOL Ltd., Tokyo, Japan) [29].

## Polymerase chain reaction (PCR)

Nutrient broth medium was used to culture *P. aeruginosa* isolates, the cultures were then incubated at 37˚C overnight. The DNA template was prepared by the boiling method. The sample is collected in a microtube and centrifuged to be concentrated. The supernatant is discarded and resuspended in the lysis buffer. This suspension is boiled for 10 minutes for cell lysis and DNA release. Finally, a brief centrifugation is performed to pellet cell debris and the supernatant containing the nucleic acid is transferred to a new microtube, and used directly in the PCR assay [30].

PCR reaction mixture was undertaken using 25 μl reaction volumes containing 12.5 μl master mix (Bioline, USA), 1 μl of each 10 $\mu mol^{-1}$ forward and reverse primers (Laboratories Midland Certified Reagent Company Inc.), 2 μl DNA template and 8.5 μl pyrogen-free water. PCR cycling using miniPCR from Amplyus (MA, USA) was performed using the conditions summarized in Table 1 as previously described [1, 25]. PCR products were analyzed using 1.5% agarose stained with Gel-Green (CA, USA) in blueGel electrophoresis unit powered with a current of 48 V obtained from blueGel built-in power supply (AC 100–240 V, 50–60 hz).

**Lipopolysaccharide SDS- polyacrylamide gel profile for *P. aeruginosa* with anticandidal activity.** Lipopolysaccharide (LPS) of the tested isolates were extracted and purified by hot aqueous- phenol method [31] and analyzed the purified material using SDS-PAGE, followed by carbohydrate-specific silver staining [32].

**GC-MS analysis of the extracted LPS.** LPS fatty acids, whole cell phospholipids, and whole cell fatty acids were derivatized to fatty acid methyl esters by methanolysis in 2 M methanolic HCl at 80˚C for 18 hrs with the addition of pentadecanoic acid as an internal standard. After the addition of an equal volume of saturated NaCl solution, the methyl esters were extracted with hexane and analyzed by gas chromatography/ mass spectrometry and the compounds were identified by ChemSpider Database [33].

**Statistical analysis.** Statistical analyses were done using SPSS version 17.0 (SPSS Inc., Chicago, IL, USA). One-Way ANOVA was employed to evaluate any significant difference between the prevalence of the tested factors among the different sources of samples. P< 0.05 indicates significant difference.

## Results and discussion

*C. albicans* and *P. aeruginosa* are commonly able to form resistant biofilms and they are often found in mixed infections, especially in cystic fibrosis and immunosuppressed patients. Many researchers have studied the way by which these organisms can co-exist with each other,

**Table 1. List of primers used in this study.**

| Gene | Primer sequence (5' 3') | PCR product (bp) | Conditions |
|---|---|---|---|
| *phzABCDEFG* | F: CCGTCGAGAAGTACATGAAT | 484 | 30 cycles of 94˚C for 30s, 60˚C for 30s, and 72˚C for 30s. |
| | R: CATAGTTCACCCCTTCCAG | | |
| *phzM* | F: AACTCCTCGCCGTAGAAC | 313 | 30 cycles of 94˚C for 30s, 60˚C for 30s, and 72˚C for 30s. |
| | R: ATAATTCGAATCTTGCTGCT | | |
| *phzs* | F: TGCGCTACATCGACCAGAG | 664 | 30 cycles of 94˚C for 30s, 60˚C for 30s, and 72˚C for 30s. |
| | R: CGGGTACTGCAGGATCAACT | | |
| *algD* | F: CGTCTGCCGCGAGATCGGCT | 313 | 30 cycles of 1 minutes at 94 ˚C,1.5 minutes at 60˚C, and 1 minutes at 72˚C |
| | R: GACCTCGACGGTCTTGCGGA | | |
| *lasB* | F: GGAATGAACGAAGCGTTCTCCGAC | 284 | 30 cycles of 1 minutes at 94˚C,1.5 minutes at 60˚C, and 1 minutes at 72 C. |
| | R: TTGGCGTCGACGAACACCTCG | | |
| *toxA* | F: CTGCGCGGGTCTATGTGCC | 270 | 30 cycles of 1 minutes at94 ˚C, 1.5 minutes at 6 ˚C, and 1 minutes at 72 ˚C. |
| | R: GATGCTGGACGGGTCGAG | | |
| *plcH* | F: GCACGTGGTCATCCTGATGC | 608 | 30 cycles of 1 minutes at 94 ˚C,1.5 minutes at 60˚C, and 1 minutes at 72˚C. |
| | R: TCCGTAGGCGTCGACGTAC | | |
| *plcN* | F: TCCGTTATCGCAACCAGCCCTACG | 481 | 30 cycles of 1 minutes at 94 ˚C,1.5 minutes at 60˚C, and 1 minutes at 72˚C |
| | R: TCGCTGTCGAGCAGGTCGAAC | | |
| *exoS* | F: CGTCGTGTTCAAGCAGATGGTGCTG | 444 | 30 cycles of 1 minutes at 94 ˚C,1.5 minutes at 60˚C, and 1 minutes at 72˚C. |
| | R: CCGAACCGCTTCACCAGGC | | |

concluding that their interaction is mainly antagonistic [17, 34, 35]. However, other studies showed that their interaction may be antagonistic or synergistic depending on environmental parameters like growth state. For example, low concentration of phenanzines will inhibit hyphal growth switching *Candida* respiration to fermentation leading to formation of ethanol which in turn caused feedback potentiation of *P. aeruginosa* in a synergistic manner [36].

Our results revealed that the tested *P. aeruginosa* showed variable antifungal activity against all tested *Candida* strains. Strains that were isolated from samples positive for *Candida* such as $P_5$, $P_8$, $P_9$, $P_{16}$, and $P_{100}$ showed variable inhibitory activity against all tested *Candida* isolates (50–100% inhibition). $P_{SA2}$ inhibited *C. glabrata* only. In addition, $P_{S1}$, $P_{S2}$, $P_{S3}$, $P_{SA1}$, $P_{82}$ and $P_{85}$ showed no or only low inhibitory activity (0–50% inhibition) according to the tested *Candida* isolates. On the other hand, three *Pseudomonas* isolates ($P_{118}$, $P_{111}$ and $P_{93}$) did not show any inhibitory activity at all against the tested *Candida* (Table 2 and Fig 1).

Light microscopic examination showed that cell free filtrate of *P. aeruginosa* inhibited germ tube formation of the tested *Candida* (*C. albicans* and *C. krusei*) (Fig 2B) in comparison to control cells (Fig 2A). Bandara, Yau [34] reported that the interaction among *P. aeruginosa* and 5 non-C. *albicans Candida* species was variable and species-specific; a finding that is in agreement with our results. Also, in keeping with our observations, variation in the effect of the tested *P. aeruginosa* on the tested *Candida* species was reported by Xu, Zeng [10]. In particular, some *P. aeruginosa* exhibited high antifungal activity and some showed partial or no activity against the tested *Candida* spp.

Strains of $P_5$, $P_8$, $P_9$ $P_{16}$ and $P_{100}$ were positive for all phenazine genes while $P_{S1}$, $P_{S2}$, $P_{S3}$, $P_{SA}$, $P_{82}$ and $P_{85}$ were positive for *phzP* and *phzM*. $P_{SA2}$ was positive for *phzP* and *phzS*. Furthermore, it was observed that $P_{118}$, $P_{111}$ and $P_{93}$ were each negative for all tested phenazine genes (Table 2 and Fig 3A).

One of the most common characteristics of *P. aeruginosa* symptoms is the presence of blue pigmentation in the pus of wounds or in the sputum, especially in case of pulmonary infections. This is due to the production of the redox active metabolite pyocyanin, a toxin. Pyocyanin production is controlled by 2 core loci (operon *phz A1B1C1D1E1F1G1* and *phz*

**Table 2. Distribution of anticandidal activity, phenazine genes and different virulence factors genes among the tested *P. aeruginosa*.**

| | *Candida* spp. | | | Phenazine genes* | | | Virulence genes* | | | | | | Inhibition of germ tube** | Source of samples |
|---|---|---|---|---|---|---|---|---|---|---|---|---|---|---|
| | *C. albicans* | *C. glabrata* | *C. krusei* | *phzP* | *phzS* | *phzM* | *toxA* | *algD* | *lasB* | *exoS* | *plcN* | *plcH* | | |
| | $C_5$/$C_9$ | $C_{T1}$/$C_{T9}$ | $C_8$/$C_{85}$ | | | | | | | | | | | |
| $P_5$ | ++++/++++ | ++++/++ | ++++/++++ | + | + | + | + | + | + | + | + | + | + | **Blood**[a] |
| $P_8$ | ++++/++++ | +++/++++ | ++++/++++ | + | + | + | + | + | + | + | - | + | + | **Blood**[a] |
| $P_9$ | ++++/++++ | ++++/++++ | ++++/++++ | + | + | + | + | + | + | + | + | + | + | **Sputum**[a] |
| $P_{16}$ | ++++/++ | +++/+++ | +++/++++ | + | + | + | + | + | + | + | + | + | + | **Sputum**[a] |
| $P_{100}$ | ++++/++++ | ++++/+++++ | ++++/++++ | + | + | + | + | + | + | + | + | + | + | **Blood**[a] |
| $P_{S1}$ | -/- | -/- | -/+ | + | - | + | + | + | + | - | - | - | + | **Wound**[b] |
| $P_{S2}$ | -/- | +/+ | -/+ | + | - | + | + | + | + | - | + | - | + | **Wound**[b] |
| $P_{S3}$ | +/+ | -/- | -/+ | + | - | + | + | + | + | + | - | + | + | **Wound**[b] |
| $P_{82}$ | -/- | -/- | -/+ | + | - | + | + | + | - | - | + | - | + | **Urine**[b] |
| $P_{85}$ | ++/++ | -/- | -/+ | + | - | + | + | + | - | + | + | + | + | **Wound**[b] |
| $P_{SA1}$ | -/- | +/- | -/+ | + | - | + | + | + | + | - | + | - | + | **Urine**[b] |
| $P_{93}$ | -/- | -/- | -/- | - | - | - | + | + | + | + | + | + | - | **Urine**[b] |
| $P_{118}$ | -/- | -/- | -/- | - | - | - | + | + | + | - | - | + | - | **Urine**[b] |
| $P_{111}$ | -/- | -/- | -/- | - | - | - | + | + | + | + | - | + | + | **Urine**[b] |
| $P_{SA2}$ | -/- | ++++/++++ | -/- | + | + | - | + | + | + | - | - | - | - | **Urine**[b] |

Anticandidal activity was interpreted as follows: -, no inhibition; +, 25% inhibition; ++, 50% inhibition; +++, 75% inhibition; ++++, 100% inhibition with respect to *Candida* culture line.

* the tested genes results were presented as–indicating the absence of the gene and + indicating the presence of the gene.

**Germ tube inhibition test was done for *C. albicans* and *C. krusei* only: +, means germ tube formation inhibition.—: means germ tubes are not inhibited.

a and b: refer to the significant differences among different source of infections in the virulence profiles in which samples carrying the same letter a or b had no significant difference while those with different letters had significant difference. P<0.05 significant value.

A2B2C2D2E2F2G2) which are responsible for the synthesis of phenazine-1-carboxylic acid (PCA) and 2 genes (*phzM* and *phzS*) encoding enzymes responsible for the conversion of phenazine-1-carboxylic acid to pyocyanin. Mavrodi, Bonsall [37] and Nowroozi, Sepahi [25] reported that the presence of both *phzM* and *phzS* are essential for pyocyanin production, a finding which may explain the decrease in the antifungal activity of strains that were negative for one of these genes or negative for both.

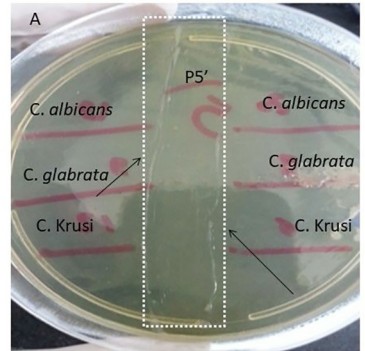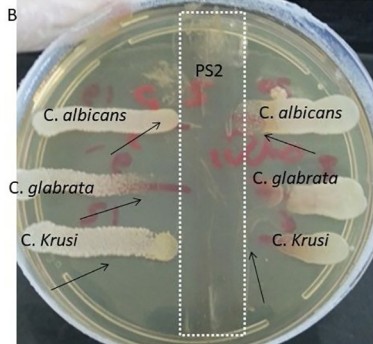

**Fig 1. Anticandidal activity of the tested *P. aeruginosa* against 6 *Candida* Spp.** A: showed complete inhibition by $P_5$ strain to 5 *Candida* strains and showed (50%) ++ inhibition of *C. glabrata*. B: showed variable activity of $P_{S2}$ strain against the tested *Candida*: no inhibition was observed against *C. albicans* and one *C. krusei*; 25% (+) inhibition against one *C. krusei* and the tested *C. glabrata*.

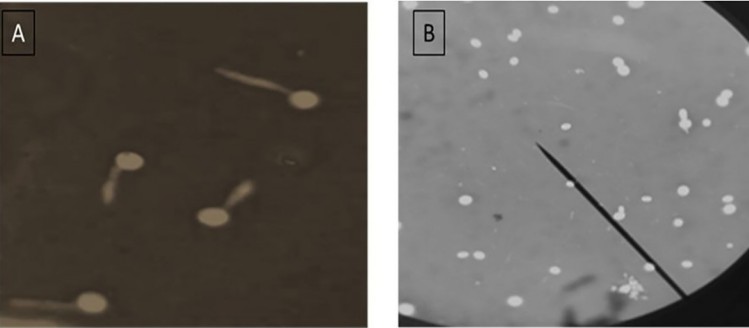

**Fig 2.** A: *C. albicans* forming germ tube after their incubation in human serum for 3 hrs. B: showed *C. albicans* with no observable germ tubes after their incubation in human serum containing *P. aeruginosa* culture cell free filtrate for 3 hrs.

All tested *P. aeruginosa* isolates were positive for *toxA* and *algD* genes, and 13 strains of the tested *P. aeruginosa* were positive for and *LasB* genes. In addition, 11 *P. aeruginosa* samples were positive for *pLcH* and 9 isolates were positive for *exoS* and *pLcN*. Table 2 and Fig 3B showed that the 5 *P. aeruginosa* showing inhibitory activity against all tested *Candida* were positive for all virulence genes except P$_8$, which was negative for *pLcN*. Many studies have reported on the important role of *P. aeruginosa* virulence factors in the development and the severity of disease and in relation to the killing effect on *Candida* Spp. [10, 38]. Caldwell, Chen

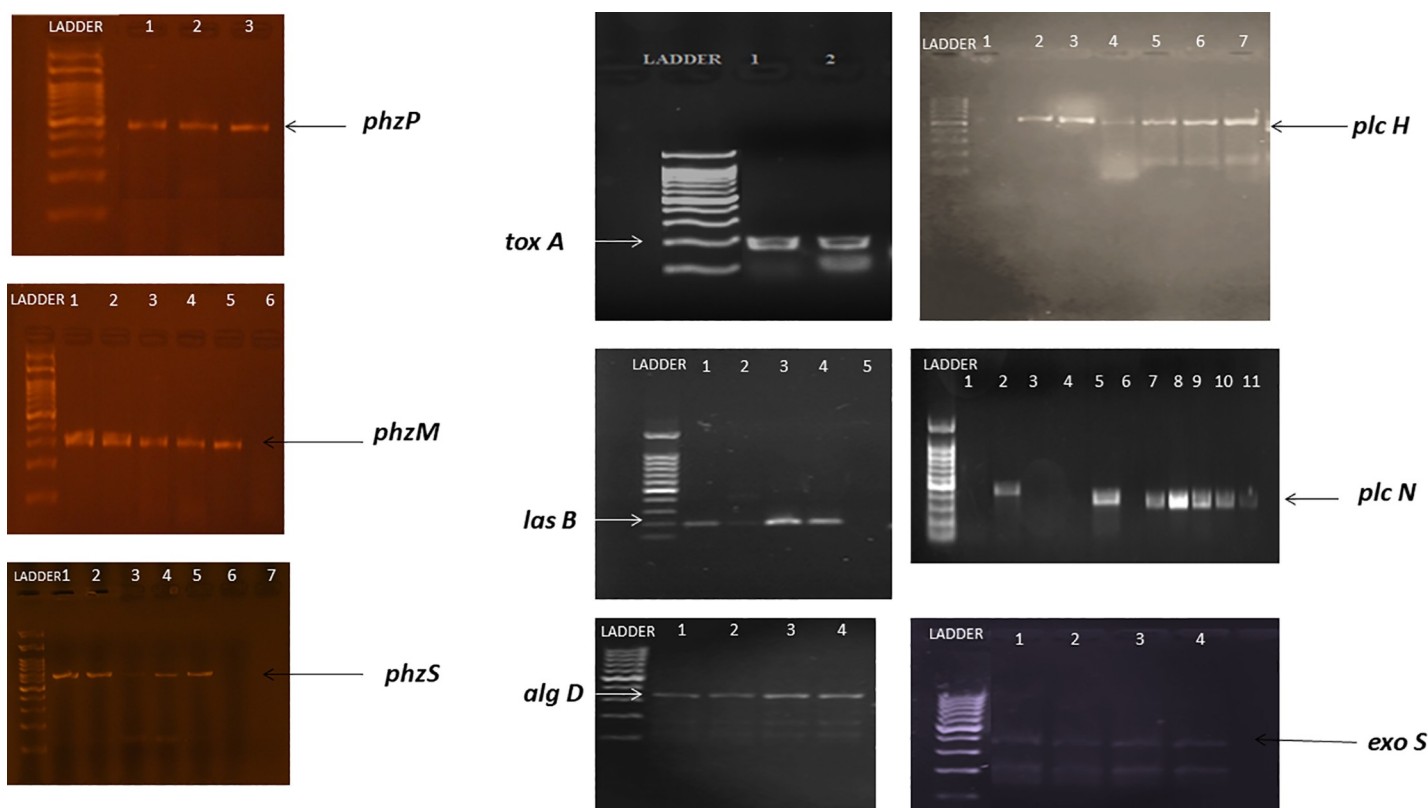

**Fig 3.** A: shows amplicons for phenazines encoding genes. B: shows amplicons for different virulence genes of the tested *P. aeruginosa*.

[35] and Taylor, Machan [39] showed that type IV pili, phospholipase C, phenazines, pyocyanin and pseudomonas quinolones secreted molecules showed inhibiting activity on *Candida* species, a finding which is in agreement with our results.

Elastase B or pseudolysin is a Zn dependent metallo-endopeptidase enzyme which is regulated by the expression of *LasB* gene controlled by quorum sensing transcription system Las and rhl [40]. This enzyme plays an important role in the invasiveness of *P. aeruginosa* during infection. It can degrade the extracellular matrix of the host cells such as elastin, vitronectin, fibronectin and collagen. In addition, it affects several components of the immune defense and some component of complement system [41, 42]. Also, *LasB* is necessary for biofilm formation [43]. The expression of *algD* gene is usually associated with the production of the mucoid colonies that protect the microorganism from the host defense factors, antibiotics and initiate colonization and adhesion to cells. Exoenzyme S is encoded by *exoS* gene which is ADP ribosyltransferase secreted by type III secretion system and this exhibits a cytotoxic effect while phospholipase C enzymes are encoded by *plcH* and *plcN* genes, enabling them to hydrolyze phospholipids [44, 45]. Phospholipase C enzymes of *P. aeruginosa* were found to recognize membrane phospholipids of eukaryotic cells. There are 2 types of phospholipases: *plcH* with hemolytic activity and *plcN* with no hemolytic activity. These enzymes were found to affect neutrophils activity, decreasing phagocytosis and increasing their survival during infection [46]. Previous studies showed that *Candida* spp. possess similar eukaryotic phospholipid membranes containing phosphatidyl choline and phosphatidyl serine. Also, Oura and Kajiwara [47] reported that sphingolipids present in *Candida* membranes play important role in the morphogenesis of *C. albicans*, pathogenesis and hyphal elongation. Hence, phospholipase enzymes production can affect *C. albicans* morphogenesis and pathogenicity.

Exotoxin S is a cytotoxin that is characterized as ADP-ribosylating enzymes for several proteins such as lipoprotein A, IgG3 and several *Ras* super family proteins. The *Ras* super family proteins are responsible for the regulation of cell proliferation, survival, and differentiation in human cells. Therefore, enzymes affecting these proteins play role in the pathogenesis of the organism during infection [48]. In *Candida albicans*, morphogenesis is stimulated by the presence of sugars, amino acids or serum via activating adenylate cyclase by different pathways [49]. Leberer, Harcus [50] reported that *Ras* super family controls the morphogenesis of *C. albicans* and the formation of true hyphae by activating adenylate cyclase after their activation by phosphorylated glucose, which indicates that *exoS* enzyme may affect hyphae formation by their effect upon *Ras* super family proteins. Additionally, ExoS has a GTPase-activating protein (GAP) activity targeting eukaryotic cells proliferation, DNA synthesis and cell morphogenesis.

Our results showed that pseudomonas strains isolated from blood and sputum were positive for the tested phenazine and virulence genes. Also, showed a powerful anticandidal activity against the tested *Candida* species. On the other hand, strains isolated from urine and swab samples showed variable activity against the tested Candida and variability in their virulence (Table 2). Hickey, Schaible [51] reported similar results. As they showed that *Pseudomonas aeruginosa* isolated from blood stream were more virulent than those isolated from peripheral sites which may be attributed to the bloodstream microenvironment and the prevalence of several proteins (LecA, RNA polymerase sigma factor 54, and proteins important for cellular metabolism and replication) stimulating the expression of many virulence factors. So, the present work confirmed a strong relationship between different virulence factors distributed among *P. aeruginosa* isolates and their ability to inhibit *Candida* hyphae (Table 2). Accordingly, *P. aeruginosa* strains showing clinical manifestations in patients such as septicemia, pneumonia, urinary infections or wound infections have potent anticandidal activity.

In the study, LPS SDS polyacrylamide gel for 4 *P. aeruginosa* with anticandidal activity and 2 isolates with no anticandidal activity was performed. The lipopolysaccharide profile revealed the

presence of Lipid A core and O-antigen repeats that appeared after performing silver staining in strains with anticandidal activity. One ($P_{111}$) of isolates with no anticandidal activity but no band for lipid A core was observed in the other strain with no anticandidal activity (**$P_{118}$**) (Fig 4).

Lipopolysaccharide profile analysis revealed the presence of Lipid A core and O-antigen repeats in *P. aeruginosa* isolates with anticandidal activity; while isolates showing no anticandidal activity showed a different LPS profile, except one ($P_{111}$) that showed a similar LPS profile to those with anticandidal activity. It has been previously reported that LPS of Gram-negative bacteria play a role in the interaction between bacteria and *Candida* spp. in mixed infections. The interaction is usually mediated by hydrophobic-electrostatic or among cell surface molecules or bacterial lipopolysaccharide and quorum sensing molecules [52, 53]. Bandara, Yau [54] and Bandara, Lam [53] showed that LPS of *P. aeruginosa*, *E. coli* and some other Gram-negative bacteria has a direct modulatory effect upon the preformed *in-vitro Candida* biofilms. In addition, LPS increases the inhibitory effect of human polymorphonuclear leukocytes to *Candida* growth. Moreover, Bandara, K Cheung [55] reported that there was a growth difference between LPS treated *Candida* and the untreated *Candida*. As they observed, a reduction

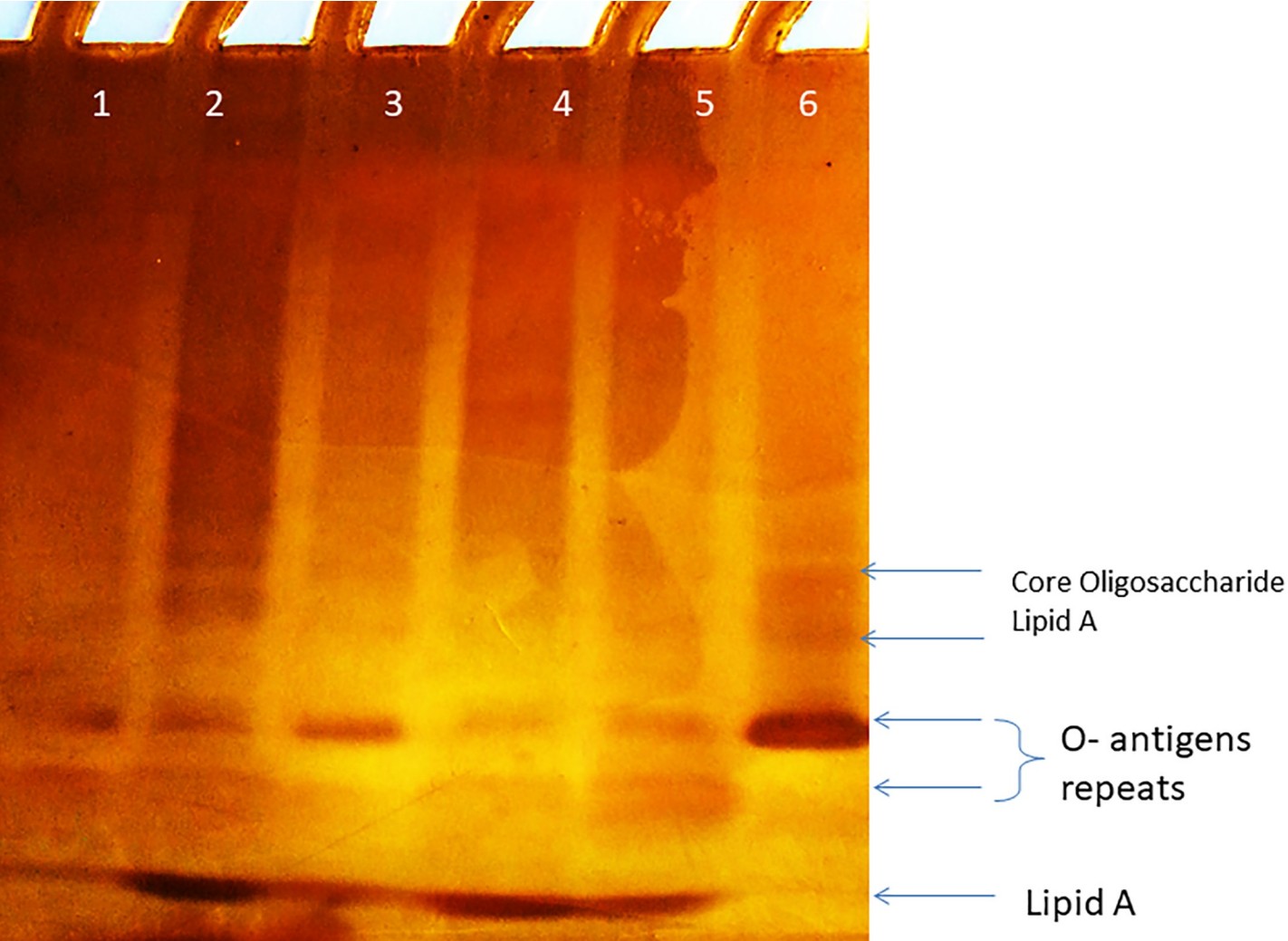

**Fig 4. LPS SDS-polyacrylamide gel profile stained by silver nitrate.** Lanes 1, 2, 3&4 ($P_5$, $P_8$, $P_9$ and $P_{16}$ isolates with anticandidal activity). Lane 5 and Lane 6 ($P_{111}$ and $P_{118 \text{ isolates}}$ with no anticandidal activity).

in the optical density in LPS treated *Candida* indicated the decrease in hyphae and yeast formation. Also, they reported that the formed biofilms appeared relative insubstantial and in the form of patchy aggregates of blastopores. Nevertheless, biofilms formed by the control strain showed well-structured filaments and dense sessile structures. In addition, the presence of LPS was found to affect ATP production by suppressing glycolysis.

Another important observation was where GC/ Mass spectra indicated that strains with anticandidal activity showed peaks that were detected in the 320–537 m/z range. These were derived primarily from the presence of rhamnolipids, recording peaks in the range of 240–320, which refers mostly to the presence of quorum sensing quinolones secreted by *P. aeruginosa*. In addition, some bioactive metabolites with antimicrobial activity were detected. Isolates with high anticandidal activity showed spectrum for some bioactive metabolites, which were known to have antimicrobial activity such as propanoic acid, 2-mercapto, methyl ester, undecanoic acid, 3- hydroxyl, methyl ester, butanoic acid, 2- methyl, heptyl ester, 1H Indole, 5-methyl-2-phenyl, quorum sensing quinolones and rhamnolipids. The presence of these compounds was found to vary among the tested strains (Figs 5–7) but it was found that

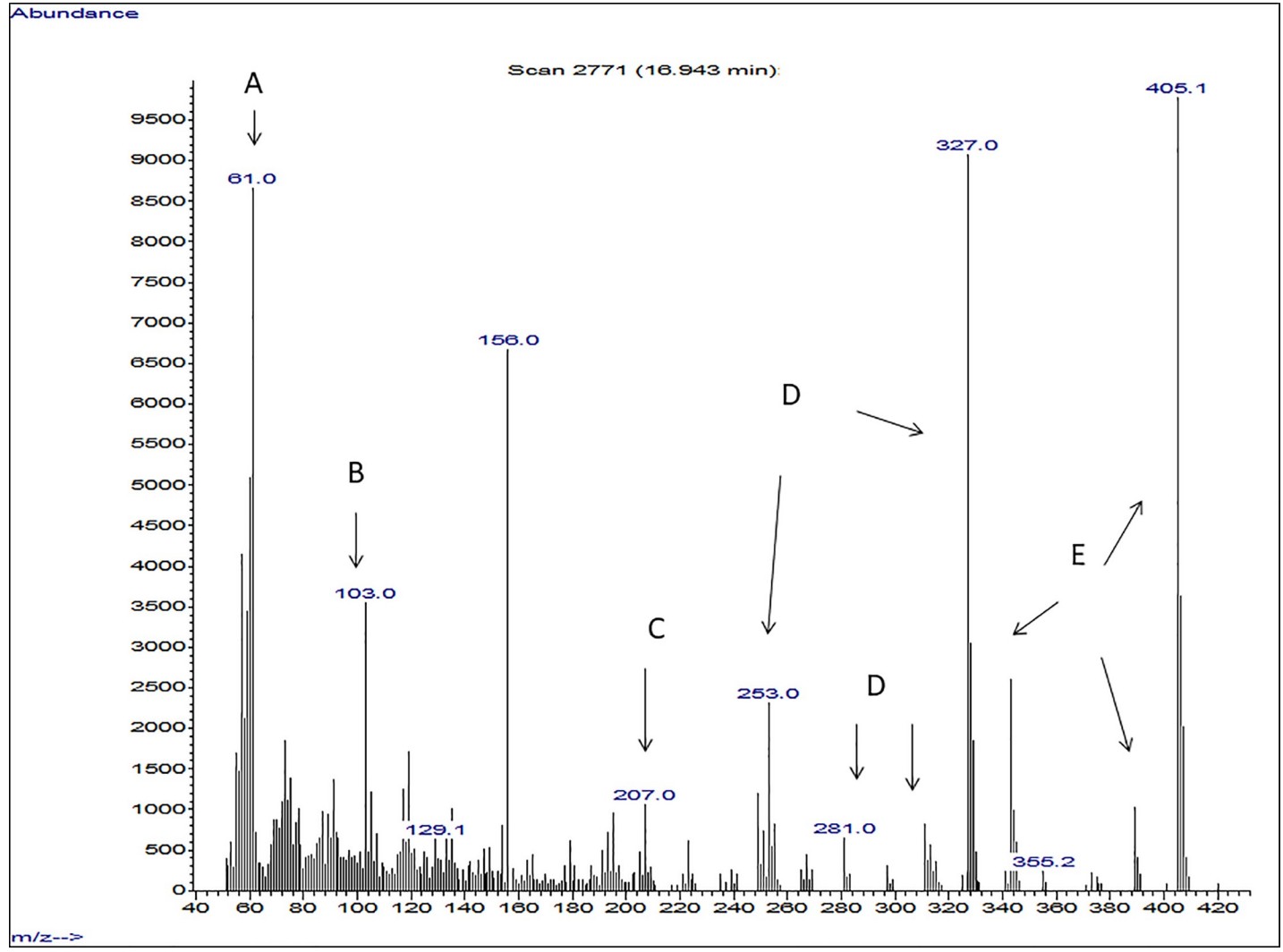

**Fig 5. Mass spectrum of the extracted LPS obtained from $P_8$ isolate.** A: Propanoic acid, 2-mercapto, methyl ester, B: Undecanoic acid, 3- hydroxyl, methyl ester, C: 1H Indole, 5-methyl-2-phenyl, D: Quorum sensing quinolones E: Rhamnolipids.

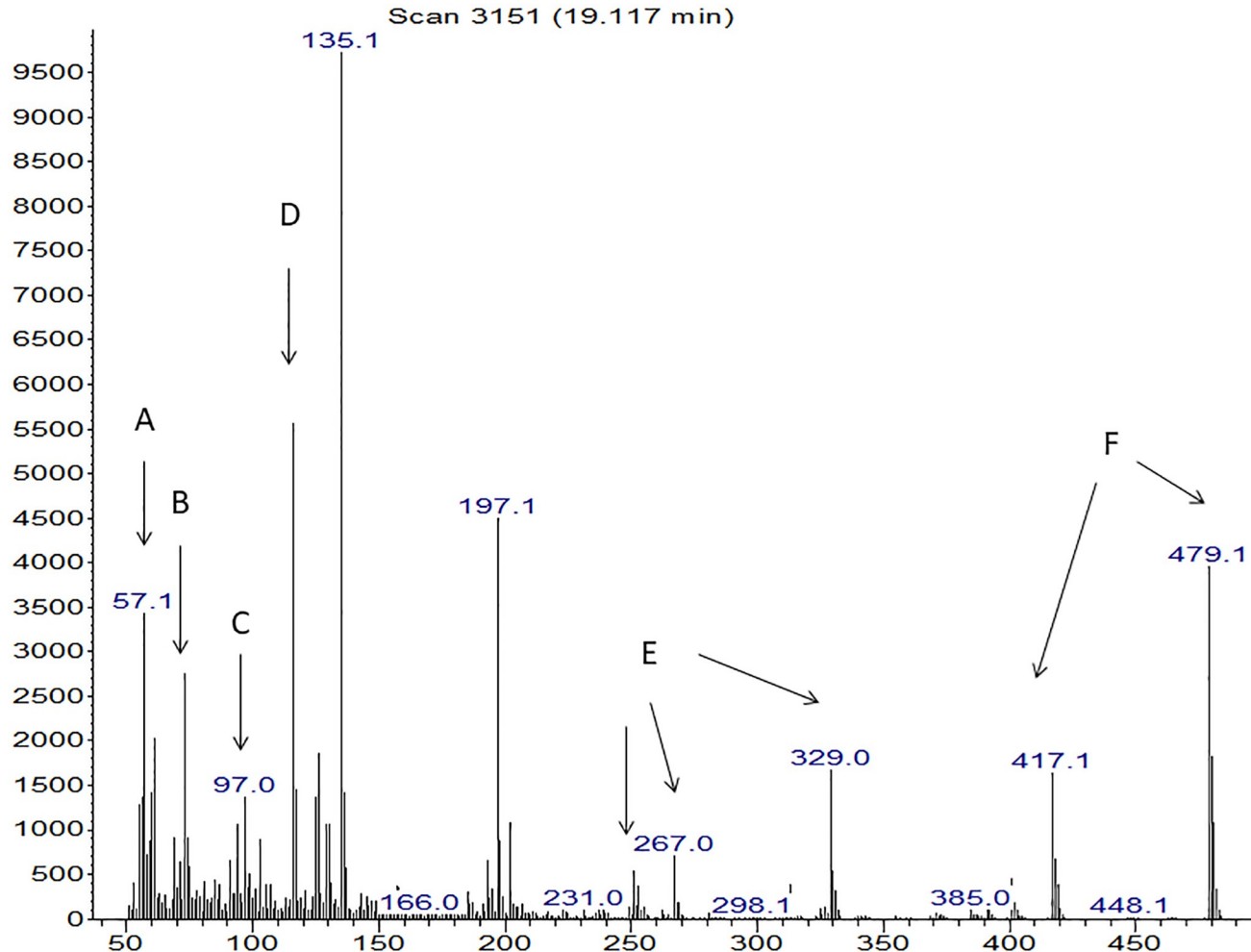

**Fig 6. Mass spectrum of the extracted LPS obtained from $P_5$ isolate.** A: Butanoic acid, 2- methyl, heptyl ester, B: Propanoic acid, 2-mercapto, methyl ester antifungal, C: Furfural or 3, 5 Dimethyl pyrazole, D: 3- isoquinolinamine, E: Quorum sensing quinolones, F: Rhamnolipids.

spectrum of $P_{SA2}$ that affect only *C. glabrata* showed peaks for Quorum sensing quinolones and rhamnolipids (Fig 8).

Several bioactive materials were identified using GC/Mass spectroscopy and the traced compounds were characterized by ChemSpider Database. Many of these compounds were found to have antibacterial, antifungal and anti-inflammatory effects according to that reported by several previous studies [56–61]. In addition, GC/Mass spectra revealed the presence of peaks in the range of 320 to 537 m/z. These are indicative of the presence of rhamnolipids. Rhamnolipids are glycolipids biosurfactants that have previously been identified and reported by many researches. Rhamnolipids were first identified in 1949 [62]. These compounds were found to be essential for biofilm formation and maturation of *P. aeruginosa*, swarming motility and the dispersion and the spread of *P. aeruginosa* from the biofilm mass. These compounds have an antagonistic activity against many types of bacteria and fungi that help in the dominance of *P. aeruginosa* during infections [62, 63]. As an example, Briard,

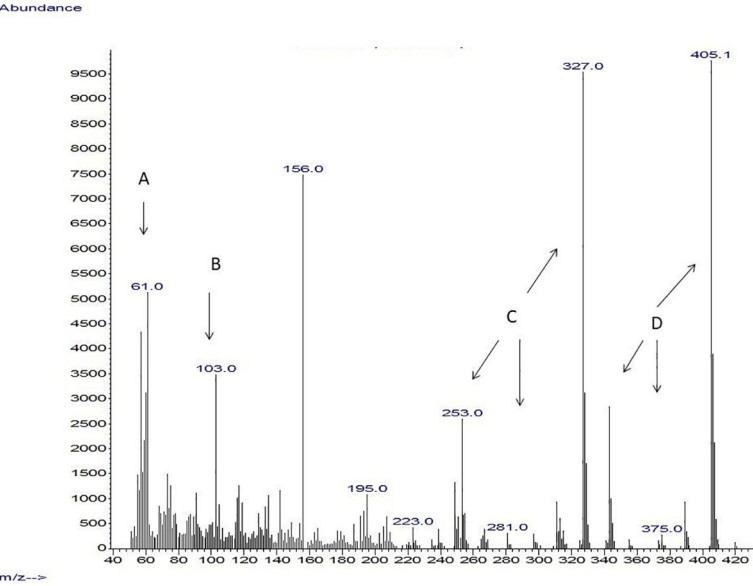

**Fig 7. Mass spectrum of the extracted LPS obtained from P$_{S3}$ isolate.** A: Propanoic acid, 2-mercapto, methyl ester (antifungal), B: Undecanoic acid, 3- hydroxyl, methyl ester, C: Quorum sensing quinolones D: Rhamnolipids.

Rasoldier [64] showed that Rhamnolipids inhibit the growth of *Aspergillus niger* by inhibiting β-1,3 glycan synthase which is essential for cell wall synthesis [65, 66]. Furthermore, strains with anticandidal activity recorded peaks in the range of 240–320 m/z, a range that indicates the presence of quorum sensing quinolones molecules. These compounds in association with acyl homoserine lactones act on the regulation of genes encoding *P. aeruginosa* virulence factors such as pyocyanin production and elastases secretion [66].

Scanning electron micrographs showed *P. aeruginosa* adhered to *Candida* cells with no hyphae and with disorganized and disrupted membranes (Fig 9A and 9B). *P. aeruginosa* tends to attach to filaments of *Candida* by their pili and killing it for gaining nutrients helping their growth [38]. Hogan and Kolter [38] showed that *P. aeruginosa* attached to *Candida* filaments in biofilm environment while *Candida* in the yeast form remains viable. Pugh and Cawson [67] reported that *Candida albicans* invasion to membranes depends on the conversion of blastospores to hyphae with phospholipase activity concentrated at the growing tip of the germ tube. So, hyphae formation is essential for the production of phospholipases. *P. aeruginosa* will attack filaments formed by *C. albicans* decreasing their ability to produce phospholipases. On the other hand, Pugh and Cawson [67] reported that non-albicans Candida are negative for phospholipase activity and *C. albicans* showed variable ability to produce phospholipases showing high quantity in blood borne infections. Clancy, Ghannoum [68] showed that non-albicans *Candida* may produce phospholipases but in very low amount (10 times much lower than *C. albicans*). According to these previous findings *P. aeruginosa* is able to inhibit the *Candida* hyphae growth as *Pseudomonas spp*. forms biofilm on *C. albicans* hyphae and kills the filamentous cells. So, it has been suggested that only the filamentous form of *C. albicans*, not the yeast form, is affected by *Pseudomonas spp*.

## Conclusion

Clinical isolates of *P. aeruginosa* showed many virulence factors, which in turn contributed to the isolates' pathogenicity and; hence, *Candida* species-specific interactions through different

**Fig 8. Mass spectrum of the extracted LPS obtained from $P_{SA2}$ isolate.** A: Quorum sensing quinolones B: Rhamnolipids.

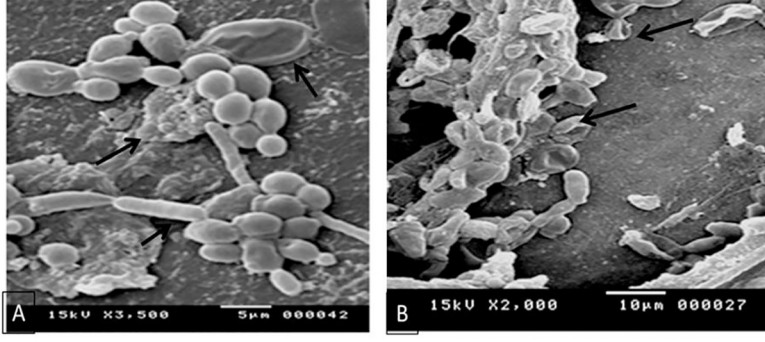

**Fig 9. Scanning electron micrograph showing *P. aeruginosa* adhered to *C. albicans* after incubating the 2 organisms together for 48 hrs.** A: *Candida* cells showed irregular cell wall and some disrupted cells with no observable hyphae and scattered *P. aeruginosa* cells. B: SEM graph showed candida cells with disorganized and disrupted cell wall after their incubation with *P. aeruginosa* culture cell free filtrate.

ways. This finding gave rise to the importance of studying microbial virulence genes and their relation to microbial interaction in polymicrobial infections and their contribution to several diseases. Also, as the currently available antifungal agents in the market are limited due to their toxicity, low effectiveness, and cost for prolonged treatment. So, it is important to develop new antifungal agents which can overcome these problems.

## Supporting information

**S1 Raw images.**
(ZIP)

## Author Contributions

**Data curation:** Rehab Mahmoud Abd El-Baky, Eman Farouk Ahmed, Zeinab Shawky Hashem.

**Formal analysis:** Eman Farouk Ahmed.

**Methodology:** Rehab Mahmoud Abd El-Baky, Sahar A. Mandour, Zeinab Shawky Hashem, Doaa Safwat Mohamed.

**Supervision:** Tim Sandle.

**Writing – original draft:** Rehab Mahmoud Abd El-Baky, Tim Sandle, Doaa Safwat Mohamed.

**Writing – review & editing:** Rehab Mahmoud Abd El-Baky, Tim Sandle.

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
