## [Decision Letter · Decision Letter 0]

6 Aug 2020

PONE-D-20-17124

Virulence profiles of some Pseudomonas aeruginosa clinical isolates and their association with the suppression of Candida growth in polymicrobial infections

PLOS ONE

Dear Dr. Abd El-Baky,

Thank you for submitting your manuscript to PLOS ONE. After careful consideration, we feel that it has merit but does not fully meet PLOS ONE’s publication criteria as it currently stands. Therefore, we invite you to submit a revised version of the manuscript that addresses the points raised during the review process.

We look forward to receiving your revised manuscript.

Kind regards,

Shankar Thangamani, DVM, PhD

Academic Editor

PLOS ONE

Journal Requirements:

2. Thank you for including the following ethics statement on the submission details page:

'"N/A" because samples are collected as part of the routine hospital laboratory

procedures.'

Please also include this information in the ethics statement in the Methods section of your manuscript.

3.PLOS ONE now requires that authors provide the original uncropped and unadjusted images underlying all blot or gel results reported in a submission’s figures or Supporting Information files. This policy and the journal’s other requirements for blot/gel reporting and figure preparation are described in detail at https://journals.plos.org/plosone/s/figures#loc-blot-and-gel-reporting-requirements and https://journals.plos.org/plosone/s/figures#loc-preparing-figures-from-image-files. When you submit your revised manuscript, please ensure that your figures adhere fully to these guidelines and provide the original underlying images for all blot or gel data reported in your submission. See the following link for instructions on providing the original image data: https://journals.plos.org/plosone/s/figures#loc-original-images-for-blots-and-gels.

4. Please ensure that you refer to Figure 1 in your text as, if accepted, production will need this reference to link the reader to the figure.

Reviewers' comments:

Reviewer's Responses to Questions

**Comments to the Author**

1. Is the manuscript technically sound, and do the data support the conclusions?

Reviewer #1: Partly

Reviewer #2: Yes

2. Has the statistical analysis been performed appropriately and rigorously? 

Reviewer #1: No

Reviewer #2: Yes

3. Have the authors made all data underlying the findings in their manuscript fully available?

Reviewer #1: Yes

Reviewer #2: No

4. Is the manuscript presented in an intelligible fashion and written in standard English?

Reviewer #1: No

Reviewer #2: Yes

5. Review Comments to the Author

Reviewer #1: This manuscript demonstrates natural phenazine compounds produced by Pseudomonas, and describe anticandidal activity against 3 different Candida spp. Then they also checked the effect on hyphae production and some virulence gene. Finally they analyzed the fatty acids by GC-MS as methyl ester derivatives. Manuscript needs more refinements in addition to several issues which need to be addressed.

1) Candida should be mentioned in ilatics in line no. 2.

2) The spelling of C. krusi is wrong, it should be C. krusei.

3) Please be accurate in writing the degree centigrade unit.

4) Candidal can be replaced by Candida in line no. 99.

5) "and" before 42°C can be removed in line no. 113.

6) It will be more appropriate to write Pseudomonas aeruginosa as P. aeruginosa instead of Ps. aeruginosa in methods section.

7) Authors can also use one more method for determining antibacterial activity.

8) Please convert rpm into g in line no. 145.

9) There is no uniformity of using units as somewhere written µl or µL. Please correct this throughout the manuscript.

10) Please mention the full name of light microscope company mentioned in line no. 152. Same should be written in line no. 167.

11) Please ensure to write either hour or hours throughout the manuscript like in line no. 153, hours is written and in line no. 156, it is written hrs.

12) figure 3B is written in line no. 228, authors have written as Figure throughout the manuscript, please be uniform.

13) Authors have used only single method for germ tube formation. There are many other hypha inducing media for studying the effects on germ tube formation. I would recommend the authors to use one more method to validate the observations.

14) Please mention the reference for method used for SEM, generally 4% glutaraldehyde is used for fixing the cell. After drying with ethanol, TMS/HMDS is used after drying and before sample mounting.

15) In section, 2.4, the method used for DNA isolation is unclear. Please mention the protocol used briefly.

16) Phospholipases enzyme are also secreted by C. albicans to degrade the host tissues (https://pubmed.ncbi.nlm.nih.gov/9467900/). So, it is unclear that how the phospholipases of one species affect the phospholipases of another species belonging to same class. Please put some more evidences to support you findings.

17) Please recheck the references no. 1 and 44.

18) Figures resolution and clarity should be ensured. Some words are blurred due to over zooming.

19) Authors have mentioned only about the antagonistic relation of P.aeruginosa and C. albicans where as this is not always the case. Please refer to this article Dhamgaye, S., Qu, Y., & Peleg, A. Y. (2016). Polymicrobial infections involving clinically relevant Gram-negative bacteria and fungi. Cellular microbiology, 18(12), 1716–1722. https://doi.org/10.1111/cmi.12674 (Studies have showed both antagonistic and synergistic interactions, depending on diverse environmental factors, timing of interaction, and growth state at the time of interaction. A lower concentration of phenazines impaired hyphal growth of C. albicans and more importantly switched fungal respiration to fermentation leading to the production of ethanol, glycerol, and acetate by C. albicans in glucose containing media (Morales et al., 2013). It was shown that C. albicans ethanol production not only influenced biofilm maturation but also promoted more phenazine production by P. aeruginosa through WspR‐dependent activation of Pel exopolysaccharide (Chen et al., 2014). The spectrum of P. aeruginosa phenazines produced was in favor of those most effective against fungal cells and led to greater production of ethanol by C. albicans , forming a feedback loop driving the polymicrobial interaction towards the protection of P. aeruginosa (Chen et al., 2014).It has now been shown that Candida colonization of the respiratory tract may promote the development of pseudomonal VAP and has been associated with the presence of multidrug‐resistant bacteria (Azoulay et al., 2006; Hamet et al., 2012). The laboratory studies characterizing the diverse interactions between Candida and Pseudomonas highlight the real complexities of their interaction, and questions still remain as to how they apply during human infection.

20) The manuscript is not thoroughly written, it need lots of language and grammar refinements.

21) There is nothing written about the statistical significance (p value) of the work.

22) The conclusion is not represented very well. Authors can rewrite this section.

23) The work is good but lacks the novelty as many previous studies have reported the interaction of both species in polymicrobial infections.

Reviewer #2: Major comments:

The authors should clarify the link between P. aeruginosa virulence and anticandidal activity in relation to clinical severity in patients. Did the strains with potent anticandidal activity show clinical manifestations in patients specific to Pseudomonas such as (pneumonia, urinary infections or wound infections) or there was no link. Please clarify and include this in discussion section. This is important since both pseudomonas and candida are opportunistic pathogens and they are part of normal flora in healthy individuals.

Minor Comments:

1-The authors should list the site of lesions for all isolated strains and clinical diseases of affected patients if there is any.

2-Conclusion should be expanded to highlight the significance of the study.

6. PLOS authors have the option to publish the peer review history of their article (what does this mean?). If published, this will include your full peer review and any attached files.

Reviewer #1: No

Reviewer #2: No

---

## [Author Response · Author response to Decision Letter 0]

1 Oct 2020

Response to reviewers were uploaded with the corrected Manuscript

---

## [Decision Letter · Decision Letter 1]

23 Nov 2020

Virulence profiles of some Pseudomonas aeruginosa clinical isolates and their association with the suppression of Candida growth in polymicrobial infections

PONE-D-20-17124R1

Dear Dr. Abd El-Baky,

We’re pleased to inform you that your manuscript has been judged scientifically suitable for publication and will be formally accepted for publication once it meets all outstanding technical requirements.

Kind regards,

Rashid Nazir

Academic Editor

PLOS ONE

Additional Editor Comments (optional):

Reviewers' comments:

Reviewer's Responses to Questions

**Comments to the Author**

1. If the authors have adequately addressed your comments raised in a previous round of review and you feel that this manuscript is now acceptable for publication, you may indicate that here to bypass the “Comments to the Author” section, enter your conflict of interest statement in the “Confidential to Editor” section, and submit your "Accept" recommendation.

Reviewer #2: (No Response)

2. Is the manuscript technically sound, and do the data support the conclusions?

Reviewer #2: Yes

3. Has the statistical analysis been performed appropriately and rigorously? 

Reviewer #2: Yes

4. Have the authors made all data underlying the findings in their manuscript fully available?

Reviewer #2: Yes

5. Is the manuscript presented in an intelligible fashion and written in standard English?

Reviewer #2: (No Response)

6. Review Comments to the Author

Reviewer #2: (No Response)

7. PLOS authors have the option to publish the peer review history of their article (what does this mean?). If published, this will include your full peer review and any attached files.

Reviewer #2: No

---

## [Editor Report · Acceptance letter]

27 Nov 2020

PONE-D-20-17124R1 

Virulence profiles of some *Pseudomonas aeruginosa* clinical isolates and their association with the suppression of *Candida* growth in polymicrobial infections 

Dear Dr. Abd El-Baky:

I'm pleased to inform you that your manuscript has been deemed suitable for publication in PLOS ONE. Congratulations! Your manuscript is now with our production department. 

Kind regards, 

on behalf of

Dr Rashid Nazir 

Academic Editor

PLOS ONE